# LEARNING ADVERSARIAL EXAMPLES WITH RIEMANNIAN GEOMETRY

## ABSTRACT

Adversarial examples, referred to as augmented data points generated by imperceptible perturbation of input samples, have recently drawn much attention. Well-crafted adversarial examples may even mislead state-of-the-art deep models to make wrong predictions easily. To alleviate this problem, many studies focus on investigating how adversarial examples can be generated and/or resisted. All the existing work handles this problem in the Euclidean space, which may however be unable to describe data geometry. In this paper, we propose a generalized framework that addresses the learning problem of adversarial examples with Riemannian geometry. Specifically, we define the local coordinate systems on Riemannian manifold, develop a novel model called Adversarial Training with Riemannian Manifold, and design a series of theory that manages to learn the adversarial examples in the Riemannian space feasibly and efficiently. The proposed work is important in that (1) it is a generalized learning methodology since Riemmanian manifold space would be degraded to the Euclidean space in a special case; (2) it is the first work to tackle the adversarial example problem tractably through the perspective of geometry; (3) from the perspective of geometry, our method leads to the steepest direction of the loss function. We also provide a series of theory showing that our proposed method can truly find the decent direction for the loss function with a comparable computational time against traditional adversarial methods. Finally, the proposed framework demonstrates superior performance to the traditional counterpart methods on benchmark data including MNIST, CIFAR-10 and SVHN.

## 1 INTRODUCTION

Recently Deep Neural Networks (DNN) achieve a big success on a wide range of challenge-able tasks, such as image classification, speech recognition, and object detection (LeCun et al., 2015)(He et al., 2017). However, recent studies have found that DNNs can be easily fooled by some special input called adversarial examples which are referred to as augmented data points generated by imperceptible perturbation of input samples (Szegedy et al., 2013)(Biggio et al., 2013)(Nguyen et al., 2015). Although such perturbation is visually imperceptible, it can lead DNNs to make incorrect predictions with high confidence.

There have been a lot of proposals studying how to generate more powerful adversarial examples, and how to build up robust networks to defend them (Goodfellow et al., 2014)(Szegedy et al., 2013). This interesting problem was first studied in (Liu & Nocedal, 1989) where the authors proposed a simple way to produce adversarial examples with L-BFGS optimization. A more powerful approach Fast Gradient Sign Method (FGSM) is later proposed in (Goodfellow et al., 2014) and further extended to a more general case with $l_p$ constraint for perturbation (Lyu et al., 2015)(Shaham et al., 2015). The multi-step variant FGSM$^k$ was proposed in (Kurakin et al., 2016) which is essentially projected gradient decent (PGD) on the negative loss. Aside from studying how to generate adversarial examples, some researchers developed methods to defend them. The adversarial training was proposed by (Goodfellow et al., 2014) (Lyu et al., 2015) which augmented the training set with adversarial examples. This method not only increases the model robustness for adversarial examples but also improves the generalization for benign samples. Some feature squeezing (Xu et al., 2017) and defensive distillation (Papernot et al., 2016) were also exploited to resist adversarial attacking. Semi-supervised version of adversarial training was developed in (Miyato et al., 2015) (Miyato et al.,

2018) (Miyato et al., 2016) called Virtual Adversarial Training (VAT), where the output distribution was smoothed by penalizing the KL-divergence between outputs of original and adversarial examples. Moreover, some researchers provided the theory and principles for adversarial examples (Fawzi et al., 2016). Furthermore, (Ma et al., 2018) have demonstrated that the adversarial examples are a dense region of pixel space instead of isolated points.

All these existing adversarial training methods simply consider the adversarial example problem in the Euclidean space with the orthogonal coordinate system. Specifically, this traditional adversarial training methods aim to solve a robust optimization problem (Lyu et al., 2015):

$$\min_{\theta} \ \mathbb{E}_{(x,y)\sim\mathcal{D}}[\max_{\epsilon}\mathcal{L}(x+\epsilon,y,\theta)] \qquad s.t. \qquad \|\epsilon\|_p \leq \sigma \qquad (1)$$

where $\mathcal{L}$ denotes a loss function and the pair of input and label $(x, y)$ is assumed to be drawn from the data distribution $\mathcal{D}$. The robust optimization problem is defined as a min-max problem with respect to the worst perturbation $\epsilon$ and the best model parameters $\theta$. The adversarial perturbation $\epsilon$ is restricted within $l_p$-ball around benign example $x$. The FGSM can be seen as a special case with $p = \infty$. Such restriction is defined in Euclidean space and the similarity between two points is measured by Minkowski distance.

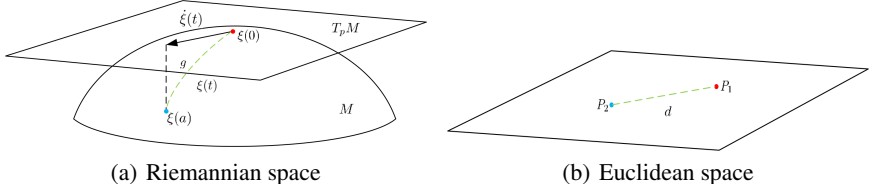

(a) Riemannian space        (b) Euclidean space

Figure 1: 1(a) shows a Riemanian space $M$, where $\xi(t)$ is the geodesic connecting two points $\xi(0)$ and $\xi(a)$. The distance between these two points is measured by the geodesic distance $g$. $T_pM$ is the tangent space of $M$ at point $\xi(0)$ and $\hat{\xi}(t)$ represents the derivative of curve function $\xi(t)$. 1(b) demonstrates the Euclidean space where the distance between two points is described by Euclidean distance $d$. Detailed notations can be seen in Section 2.

However, data points may be in practice attached on a geometric manifold which cannot be appropriately described with Euclidean coordinate system. In this case, the Euclidean metric would be not rational. Moreover, existing adversarial training methods usually search the worst perturbation through the gradient of loss function with respect to $x$, since the gradient is considered as the steepest direction. However, in a geometric manifold, particularly in Riemannian space, the gradient of a loss function unnecessarily presents the steepest direction. Figure 1 illustrates the difference between a Riemannian space and the Euclidean space, where the detailed mathematic notation can be seen in Section 2. Clearly, In this figure, assuming that the data are attached in the manifold as defined in Figure 1(a), the Euclidean distance may not appropriately reflect the true distance between two points (e.g., $\xi(0)$ and $\xi(a)$).

In this paper, we extend the traditional adversarial problem to Riemannian space and propose a novel adversarial method called Adversarial Training with Riemannian Manifold (ATRM). ATRM is regarded as a generalized framework in that Riemannian space contains the Euclidean space as a special case. In more details, we start with defining the local coordinate system and Riemannian metric tensor to evaluate the similarity between two points in Riemannian space. We then propose to restrict the adversarial perturbation within $l_p$-ball around natural examples $x$ on Riemannian manifold, and develop a framework to search the worst perturbation through the generalized trust region method. Our proposed method is to solve the adversarial problem from the perspective of geometry which is similar to Natural Gradient methods (Amari, 1998), however, our method is implemented on input space instead of parameter space of DNNs.

We list the main contributions of this paper as follows: 1) To our best knowledge, this is the first work to tackle the adversarial example problem through the perspective of geometry. 2) We study the adversarial example in the more generalized Riemannian space of which Euclidean space is a special case. 3) Our method considers the curvature information of the loss function which can be viewed as the second order method, enabling a more accurate direction of adversarial perturbation. Importantly, from the perspective of geometry, our method leads to the steepest direction of loss

function in Riemannian space. 4) We also provide a series of theory showing that our proposed method can truly find the decent direction for the loss function with a comparable computational time against traditional adversarial method (one more backward backward prorogation).

## 2 MAIN METHODOLOGY

In this section, we first study how to generate adversarial examples within $l_2$-ball in Riemannian Manifold. We then define the local metric tensor on the manifold and generate the adversarial examples using our proposed ATRM with $l_2$-ball restriction. Next, we extend our proposed method to $l_p$-ball restriction on Riemannian manifold. Finally, we propose the adversarial training method called ATRM and also provide theories showing that ATRM can find the decent direction of the loss.

### 2.1 RIEMANNIAN GEOMETRY

In this subsection, we first introduce the mathematical notations and some background of geometry. The Einstein notation is used in this subsection. The Riemannian Manifold is defined as Definition 2.1:

**Definition 2.1.** (Riemannian Manifold (Walschap, 2012)) In differential geometry, a Riemannian manifold $(M, g)$ is a real smooth manifold $M$ equipped with inner product in tangent space $T_p M$ at each point $p$ varying smoothly on $M$, defined by positive definite metric tensor $g_p$.

**Theorem 2.1.** *(Geodesic (Amari, 2016)) A curve that connects two points by a minimal distance is a geodesic under the Levi-Civita connection, where the length of a curve $c = \xi(t)$ connecting $\xi(0)$ and $\xi(a)$ is given by*

$$l = \int_0^a \sqrt{g_{ij}(t)\dot{\xi}^i(t)\dot{\xi}^j(t)} \ dt \tag{2}$$

*where $g_{ij}(t)$ denotes the metric tensor at point $\xi(t)$.*

The geodesic can be calculated with Theorem 2.1, which measures the distance of two points on Riemannian manifold. It can be seen as an extension of the Euclidian distance in Euclidian space.

**Lemma 2.2.** *Let $\xi(0)$ and $\xi(a)$ be two close points on Riemannian manifold, where $\xi(t)$ is a shortest curve connecting these two points. Then, the distance between these two points can be computed by:*

$$ds^2 = g_{ij}(t)d\theta^i d\theta^j \tag{3}$$

*where $d\theta^i = \dot{\xi}(t)^i dt$ is the $i^{th}$ component of small vector $d\theta$ and $g_{ij}(t)$ is the metric tensor (proof can be seen in Appendix **D**).*

We can also rewrite $ds^2$ in form of inner production:

$$ds^2 = \langle d\theta^i e_i, d\theta^j e_j \rangle = \langle e_i, e_j \rangle d\theta^i d\theta^j \tag{4}$$

where basis vectors $\{e_i\}$ is the set of tangent vectors along the coordinate curves. Combining these two equations, we have:

$$g_{ij} = \langle e_i, e_j \rangle \tag{5}$$

Therefore, the metric tensor $G = (g_{ij})$ is the inner product of basis vectors. In the case of Euclidean space (orthonormal coordinate system), the metric tensor is:

$$g_{ij} = \delta_{ij} = \begin{cases} 1 & i{=}j \\ 0 & \text{otherwise} \end{cases} \tag{6}$$

where $\delta_{ij}$ represents the Kronecker delta. The Euclidean space can be viewed as a special case of Riemannian space with the identity metric tensor. For traditional adversarial training, the data are assumed to be in Euclidean space with the identity metric tensor. In this paper, we consider a more general case that the data are attached on the Riemannian manifold with positive definite metric tensor varying smoothly on the manifold. Figure 1 illustrates the difference between the two spaces.

## 2.2 Adversarial Perturbation within $l_2$-ball on Manifold

In this subsection, we introduce in details how we exploit a generalized trust region method to search the adversarial perturbation on the Riemannian manifold.

To search the adversarial examples on the Riemannian manifold feasibly, we propose to solve the inner optimization problem of (1) with the generalized trust region method:

$$\epsilon = \arg\max_{\epsilon} \ \mathcal{T}_k\{\mathcal{L}, x, \epsilon\} \qquad s.t. \qquad d(x, x + \epsilon) \leq \sigma \tag{7}$$

where $\epsilon$ is a small perturbation and $\sigma$ is a small value. $\mathcal{T}_k\{\mathcal{L}, x, \epsilon\}$ denotes the $k^{th}$ order Taylor expansion of function $\mathcal{L}(.)$ around $x$ evaluated at $x + \epsilon$. Here $k$ can be set to 1 or 2. $d(x, x + \epsilon)$ represents the distance between two points $x$ and $x + \epsilon$. In this paper, we follow the setting of previous research (Lyu et al., 2015) (Shaham et al., 2015) and provide the $1^{th}$ Taylor expansion for the loss function $\mathcal{L}(.)$. We define the $d(.)$ geodesic distance between two points on Riemannian manifold. Since the perturbation $\epsilon$ is very small, by applying Lemma 2.2, (7) can be reformulated as:

$$\epsilon = \arg\max_{\epsilon} \ \mathcal{L}(x) + \nabla_x \mathcal{L}^T \epsilon \qquad s.t. \qquad g_{ij}(x)\epsilon^i \epsilon^j \leq \sigma^2 \tag{8}$$

where $G = (g_{ij}(x))$ denotes the metric tensor at the point $x$ on data manifold. In (7), since geodesic $d(x, x + \epsilon) \leq \sigma$, the geodesic can be evaluated by (3). It can be proved that we can solve the problem (8) with Lagrangian multiplier method and the worst perturbation can be given as:

$$\epsilon \propto G^{-1} \nabla_x \mathcal{L} \tag{9}$$

The detailed proof of (9) can be seen in Appendix **A**.

## 2.3 Defining Metric Tensor

In the previous subsection, we have derived that the direction of the worst perturbation is relevant to metric tensor and the first derivative of $\mathcal{L}$ with respect to $x$. It is easy to evaluate $\nabla_x \mathcal{L}$ in DNNs through back propagation. It is then very crucial to define the Riemannian manifold for input and the metric tensor $G$. Since, the adversarial examples are closely related to loss function and classification boundary, we can define the Riemannian manifold and metric tensor from loss function. Specifically, We first take the $n^{th}$ order Taylor expansion of $\mathcal{L}$ around $x$ evaluated at $x + \epsilon$ ($n$ is an extremely large value):

$$\mathcal{L}(x + \epsilon) \cong \mathcal{L}(x) + \frac{\mathcal{L}_x^{(1)}(x)}{1!}\epsilon + \frac{\mathcal{L}_x^{(2)}(x)}{2!}\epsilon^2 + ... + \frac{\mathcal{L}_x^{(n)}(x)}{n!}\epsilon^n$$
$$= \mathcal{L}(x) + \frac{\mathcal{L}_x^{(1)}(x)}{1!}\epsilon + m(x) \tag{10}$$

where $\mathcal{L}_x^{(n)}(x)$ denotes the $n^{th}$ order derivative of $\mathcal{L}$ with respect to $x$ and

$$m(x) = \frac{\mathcal{L}_x^{(2)}(x)}{2!}\epsilon^2 + o(x) \tag{11}$$

The loss function $\mathcal{L}(x + \epsilon)$ can be written as summation of its $1^{th}$ order Taylor expansion and the other components of Taylor series $m(x)$. The $1^{th}$ order Taylor expansion is more reliable when $|m(x)|$ is small enough. Naturally, the restriction can be defined as:

$$|m(x)| \cong \left|\frac{\mathcal{L}_x^{(2)}(x)}{2!}\epsilon^2\right| = \frac{1}{2}|\epsilon^T H \epsilon| \leq \sigma^2 \tag{12}$$

However, it is difficult to deal with the constraint (12) in the optimization problem. We can simplify it with its upper bound by using Lemma 2.3.

**Lemma 2.3.** *Assume $H$ be a symmetric square matrix in $\mathbb{R}^{n \times n}$ and $r \in \mathbb{R}^n$ be a vector. Then we have $|r^T H r| \leq r^T |H| r$ and $|H|$ represents the matrix with taking the absolute value of each eigenvalue of $H$ (proof can be seen in Appendix **E**).*

Based on Lemma 2.3, we can have

$$\frac{1}{2}|\epsilon^T H \epsilon| \leq \frac{1}{2}\epsilon^T |H| \epsilon \leq \sigma^2 \tag{13}$$

where $H$ is the Hessian matrix of $\mathcal{L}$ with respect to $x$ and $\sigma$ is a small value. When the loss function is locally convex with respect to $x$, the Hessian matrix $H$ is positive semidefinite matrix and the absolute Hessian matrix is the same as the Hessian matrix. When the loss function is not locally convex, the non-negative eigenvalues of absolute Hessian matrix keep the same with the Hessian matrix while the negative eigenvalues are changed to positive ones. The curvature information is partially kept. (13) can be viewed as the distance between two close points $x$ and $x + \epsilon$ on the Riemannian manifold with metric tensor $|H|$.

Comparing (13) with constraint of (8), we can define the metric tensor as $|H|$. Substituting $|H|$ in (8), we can easily get the worst perturbation:

$$\epsilon \propto |H|^{-1}\nabla_x \mathcal{L} \tag{14}$$

In contrast to the traditional adversarial training methods, the metric tensor $|H|$ of our method involves the curvature information of the loss function which can be seen as the second order method. Through the perspective of geometry, the direction of gradient is not guaranteed to be steepest in Riemannian space, however, the metric tensor adjusts it to the steepest one as illustrated in Figure 2. Note that, for DNNs, it is easy to evaluate the Hessian matrix of $\mathcal{L}$ with respect to input $x$ by back propagation.

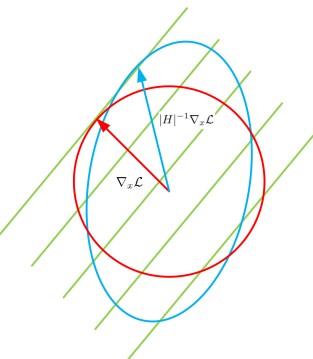

Figure 2: The red circle denotes the region of $\|\epsilon\|_2 \leq \sigma$, while the red arrow illustrates the direction of gradient. The blue ellipse shows the region of $\epsilon^T |H| \epsilon \leq \sigma^2$, the blue arrow represents the direction found by the proposed method, and green lines present the contour lines. The steepest direction is given by the gradient of the loss function, which is orthogonal to contour lines only when an orthonormal coordinate system is used in Euclidean space. In the Riemannian space, the steepest direction is not guaranteed to be orthogonal to contour lines. However, adjusted with the metric tensor $|H|^{-1}$, the direction of gradient can be approximate to the steepest direction. The graph is better viewed in color.

### 2.4 ADVERSARIAL PERTURBATION WITHIN $l_p$-BALL ON MANIFOLD

In the previous subsections, we have calculated the worst perturbation within $l_2$-ball on Riemannian manifold. Similar to traditional adversarial training methods, we can also extend our method to $l_p$-ball on manifold. First, we introduce Lemma 2.4 before we proceed to the case of $l_p$-ball.

**Lemma 2.4.** *Let $A$ a real symmetric positive definite matrix in $\mathbb{R}^n \times \mathbb{R}^n$. Then we have a unique positive definite matrix $S$ in $\mathbb{R}^n \times \mathbb{R}^n$ so that $A = S^2$ (proof can be seen in Appendix **F**).*

Using Lemma 2.4, we reformulate the constraint of (13)

$$\epsilon^T |H| \epsilon = \epsilon^T S S^T \epsilon = (\epsilon^T S)^2 \leq \sigma^2 \tag{15}$$

where $S$ is a positive definite matrix which we call a transformation matrix. Then we can easily extend (15) to $l_p$-ball on manifold:

$$\|\epsilon^T S\|_p = (\sum_i |\epsilon^T S|_i^p)^{1/p} \leq \sigma \tag{16}$$

Substituting (16) with the constraint of (8), we can easily evaluate the corresponding worst perturbation (details are provided in Appendix **B**):

$$\epsilon = \sigma sign(\nabla \mathcal{L}^T S^{-1})(\frac{|\nabla \mathcal{L}^T S^{-1}|}{\|\nabla \mathcal{L}^T S^{-1}\|_{p^*}})^{\frac{1}{p-1}} S^{-1} \tag{17}$$

where $p^*$ is the dual of $p$, i.e.,$\frac{1}{p^*} + \frac{1}{p} = 1$. Clearly, when $p = 2$, the worst perturbation is reduced to (14) which is the case of the perturbation within $l_2$-ball on manifold. When $p = \infty$, our method reduces to the generalized FGSM:

$$\epsilon = \sigma \lim_{p \to \infty} sign(\nabla \mathcal{L}^T S^{-1})(\frac{|\nabla \mathcal{L}^T S^{-1}|}{\|\nabla \mathcal{L}^T S^{-1}\|_{p^*}})^{\frac{1}{p-1}} S^{-1} = \sigma sign(\nabla \mathcal{L}^T S^{-1}) S^{-1} \tag{18}$$

Though we can evaluate the adversarial perturbation for any $l_p$-ball on Riemannan manifold through (17). In this paper, we focus on the constraint of $l_2$-ball and will develop the corresponding adversarial training method to improve the performance of DNNs in the next subsection.

## 2.5 ADVERSARIAL TRAINING METHOD WITH RIEMANNIAN MANIFOLD

After studying the adversarial examples on manifold, we now consider to design an optimization method to improve the DNNs using the theory in the previous subsections. We first define the overall optimization as the robust optimization problem in a way similar to the traditional adversarial training:

$$\min_\theta \quad \mathbb{E}_{(x,y) \backsim \mathcal{D}}[\max_\epsilon \mathcal{L}(x + \epsilon, y, \theta)] \qquad s.t. \quad \|\epsilon^T S\|_p \leq \sigma \tag{19}$$

In this paper, we focus on the $l_2$-ball constraint, while it is easily extended to the $l_p$-ball constraint. In the case of $l_2$, we can reduce the constraint in (19) to $\epsilon^T |H| \epsilon \leq \sigma$. To optimize this problem, we can first solve the inner optimization problem then followed by the outer one. We then repeat this process until it converges. The whole process is shown as Algorithm 1. On the other hand, Algorithm 2 demonstrates the function of approximating $|H|^{-1} \nabla_x \mathcal{L}(y_i, x_i, \theta)$. Hessian matrix may require a large amount of computation. In Algorithm 1, we approximate $|H|^{-1} \nabla_x \mathcal{L}(y_i, x_i, \theta)$ with the first derivative of loss function with respect to input. Specifically, we simply modify the Limited-memory BroydenCFletcherCGoldfarbCShanno (L-BFGS) method. More details can be seen in Appendix **C**.

---

**Algorithm 1** Framework of ensemble learning for our system.

1: **for** number of training iterations **do**
2:     Sample a batch of labeled data $(x_i, y_i)$ with size $N$.
3:     **for** $i$ in $1...N$ **do**
4:         $d_i \leftarrow \overline{|H|^{-1} \nabla_x \mathcal{L}(y_i, x_i, \theta)}$
5:         $\epsilon_{adv}^i = \xi d_i$
6:     **end for**
7:     Update the parameters of neural network with stochastic gradient:
8:
9:     $-\nabla_\theta \frac{1}{N} \sum_{i=1}^N log \mathcal{L}(y_i, x_i + \epsilon_{adv}^i, \theta)$
10: **end for**

---

In Algorithm 1, $\overline{|H|^{-1} \nabla_x \mathcal{L}(y_i, x_i, \theta)}$ represents the normalization of $|H|^{-1} \nabla_x \mathcal{L}(y_i, x_i, \theta)$. We now provide the theoretical analysis showing that our proposed method truly offers the decent direction for the optimization problem as (Madry et al., 2017). In order to do this, we first present Theorem 2.5.

---

**Algorithm 2** Approximation for $|H|^{-1}\nabla_x\mathcal{L}(y_i, x_i, \theta)$.

1: **function** APPROHD$(y_i, x_i, \theta, \zeta)$
2:      Give $\zeta$ very small value
3:      $g_0 = \nabla_x\mathcal{L}(y_i, x_i, \theta)$
4:      $g_1 = \nabla_x\mathcal{L}(y_i, x_i + \zeta g_0, \theta)$
5:      $y = g_1 - g_0$
6:      $s = \zeta g_0$
7:      $\rho = \frac{1}{y^T s}$
8:      $\alpha = \rho s^T g_1$
9:      $q = g_1 - \alpha y$
10:     $r_0 = q_0$
11:     $\beta = \rho y^T r_0$
12:     $r_1 = r_0 + (\alpha - \beta)s$
13:     **return** $r_1$
14: **end function**

---

**Theorem 2.5.** *(Danskin). Let $S$ be nonempty compact topological space and $g : \mathbb{R} \times S \to \mathbb{R}$ such that $g(., \delta)$ is differentiable for every $\delta \in S$ and $\nabla_\theta g(\theta, \delta)$ is continuous on $\mathbb{R}^n \times S$. Also assume $\delta^*(\theta) = \{\delta \in \arg\max_{\delta \in S} g(\theta, \delta)\}$. Then the max-function $\phi(\theta) = \max_{\delta \in S} g(\theta, \delta)$ is locally Lipschitz continuous, directionally differentiable, and its directional derivatives satisfy: $\phi'(\theta) = \sup_{\delta \in \delta^*(\theta)} h^T \nabla g(\theta, \delta)$ In particular, if for some $\theta \in \mathbb{R}^n$ the set $\delta^*(\theta) = \{\delta_\theta^*\}$ is a singleton, the max-function is differentiable at $\theta$ and $\nabla\phi(\theta) = \nabla_\theta g(\theta, \delta_\theta^*)$.*

This theorem states that the gradients of $\phi(\theta)$ are local objects and the gradients are locally the same as that of $g(\theta, \delta_\theta^*)$. With Theorem 2.5, we describe the theory showing that our proposed optimization method truly offers the decent direction:

**Lemma 2.6.** *Let $\hat{\delta} \in S$ be a maximizer of $\max_\delta \mathcal{L}(\theta, x + \theta, y)$. Then, we have that $-\nabla_{theta}\mathcal{L}(\theta, x + \hat{\delta}, y)$ is a decent direction for $\phi(\theta) = \max_{\delta \in S} \mathcal{L}(\theta, x + \delta, y)$.*

*Proof.* We apply Theorem 2.5 that $g(\theta, \delta) := \mathcal{L}(\theta, x + \delta, y)$ and $S = B_p(\sigma)$, which is defined as the $l_p$-ball with radius $\sigma$ on Riemannian manifold. The directional derivative in the direction of $h = \nabla_\theta\mathcal{L}(\theta, x + \hat{\delta}, y)$ satisfies:

$$\phi'(\theta, h) = \sup_{\delta \in \hat{\delta}(\theta)} h^T \nabla_\theta L(\theta, x + \delta.y) \geq h^T h = \|\nabla_\theta\mathcal{L}(\theta, x + \hat{\delta}, y)\|_2^2 \geq 0 \tag{20}$$

$\square$

### 2.6 COMPUTATIONAL ANALYSIS

Compared with traditional adversarial training methods, our proposed ATRM need compute the Hessian matrix of the loss function with respect to input additionally. It may cost extra computation to calculate the Hessian matrix. Nonetheless, we exploit in this paper the first-order derivative of the loss function to approximate the product of Hessian matrix and the vector, which is shown in Algorithm 2. Therefore, our proposed method requires backward propagation three times and forward prorogation once. Specifically, the first backward prorogation is used to approximate the Hessian matrix, the second one is used to evaluate the adversarial perturbation, and the last time is to update the parameters of DNNs. In contrast to traditional adversarial training methods, our proposed method need merely one additional backward propagation which is acceptable in practice.

## 3 EXPERIMENT

To validate the efficacy of our proposed method, we conduct a series of experiments on benchmark data including MNIST, CIFAR-10, and SVHN. In these experiments, we compare our proposed ATRM with other competitive methods. For MNIST datasets we use the same baseline

as (Rasmus et al., 2015). The same base structure called 'conv-large' is used on dataset CIFAR-10 and SVHN, which follows (Miyato et al., 2018). Specifically, 'conv-large' means a large convolutional neural network with seven convolutional layers and three fully-connected layers with dropout where the size of all the convolutional kernels is $3 \times 3$.

We first implement our proposed ATRM on handwriting dataset MNIST. Since there is no previous adversarial research on this baseline model on this dataset, we conduct the experiment on two methods (adversarial training with $l_\infty$ and $l_2$ constraint) for comparison. The model is trained with $60,000$ labeled samples without any data augmentation and is tested with $10,000$ samples. We train the model with a batch size of 32 and the maximum 500 epochs. There are two hyper parameters $\{\zeta, \xi\}$ in Algorithm 1 and Algorithm 2. We set $\zeta$ to a very small value, which is $10^{-6}$ in this paper. We tune the value of $\xi$ in the range of $\{0.1, 0.2, 0.5, 1, 2, 5, 10\}$. We use the set of hyper parameters that achieved the best performance on the validation set of size $5,000$, selected randomly from the pool of training samples of size $60,000$.

Table 1 lists the performance of various methods including our proposed ATRM and other competitive methods. Except for ATRM, we also conduct the same experiment for the baseline model and traditional adversarial training methods with $l_2$-ball and $l_\infty$-ball constraint. We conduct the experiment with this same setting for five times and calculate the mean and standard deviation. As observed, our proposed ATRM demonstrates the best performance. In particular, all the adversarial methods achieve remarkably good performance, while the ATRM shows a further improvement. This validates the advantages of our proposed method over the other methods, especially the other traditional adversarial methods.

Table 1: Test performance on MNIST

| Method | MNIST Test error rate(%) |
|---|---|
| SVM | 1.40 |
| Dropout (Srivastava et al., 2014) | 1.05 |
| Ladder networks (Rasmus et al., 2015) | $0.57 \pm 0.02$ |
| VAT (Miyato et al., 2018) | 0.72 |
| RPT (Miyato et al., 2018) | 0.82 |
| Baseline (Rasmus et al., 2015) | 0.32 |
| Baseline+$l_\infty$ adversarial training | $0.30 \pm 0.013$ |
| Baseline+$l_2$ adversarial training | $0.26 \pm 0.019$ |
| ATRM | $\mathbf{0.22} \pm 0.016$ |

For CIFAR-10, we train our proposed model ATRM with $50,000$ labeled samples with data augmentation as conducted in (Miyato et al., 2018) (translation and horizontal flip). The test dataset of CIFAR-10 involves $10,000$ samples. Similar to the experiment on MNIST, we set $\zeta$ to a small vaule $10^{-6}$, and tune the value of $\xi$ in the range of $\{1, 2, 5, 8, 10\}$. We run the experiments for five times and report the average performance and corresponding standard deviation.

Table 2 summarizes the results of different methods on CIFAR-10 dataset. In this experiment, we intentionally compare our proposed method ATRM with the other two very deep models, i.e., the densely connected network (DenseNet) with 190 layers and very deep residual network (ResNet) with $1,001$ layers. Overall, our proposed method demonstrates competitive performance. Though not as good as the very deep networks DenseNet and ResNet, the proposed ATRM shows superior performance to those adversarial learning methods and all the other remaining approaches. Once again, this shows that adversarial learning should be conducted on the geometric manifold rather than the traditional Euclidean space.

The Street View House Numbers (SVHN) dataset contains $32 \times 32$ colored digit images. There are $73,257$ images in the training set, $26,032$ images in the test set, and $531,131$ images for additional training. We train our model using the same setting as (Huang et al., 2017). As can be clearly observed in Table 3, our method demonstrates the best performance. It is even much better than the very deep networks (DenseNet and ResNet). More importantly, our method achieves an obvious improvement compared with the traditional adversarial training algorithms.

Table 2: Test performance on CIFAR-10

| Method | CIFAR-10 Test error rate(%) |
|---|---|
| Network in Network (Lin et al., 2013) | 8.81 |
| All-CNN (Springenberg et al., 2014) | 7.25 |
| Deeply Supervised Net (Lee et al., 2015) | 7.97 |
| Highway Network (Srivastava et al., 2015) | 7.72 |
| RPT (Miyato et al., 2018) | $6.25 \pm 0.04$ |
| ResNet (1,001 layers) (He et al., 2016) | $4.62 \pm 0.2$ |
| DenseNet (190 layers) (Huang et al., 2017) | **3.46** |
| Baseline (Miyato et al., 2018) | $6.76 \pm 0.07$ |
| VAT (Miyato et al., 2018) | $5.81 \pm 0.02$ |
| Baseline+$l_\infty$ adversarial training | $6.35 \pm 0.03$ |
| Baseline+$l_2$ adversarial training | $5.82 \pm 0.02$ |
| ATRM | $5.35 \pm 0.03$ |

Table 3: Test performance on SVHN

| Method | SVHN Test error rate(%) |
|---|---|
| Network in Network (Lin et al., 2013) | 2.35 |
| Deeply Supervised Net (Lee et al., 2015) | 1.92 |
| ResNet (110 layers) (He et al., 2016) | 2.01 |
| DenseNet (250 layers) (Huang et al., 2017) | 1.74 |
| Baseline | $2.09 \pm 0.06$ |
| Baseline+$l_\infty$ adversarial training | $1.95 \pm 0.05$ |
| Baseline+$l_2$ adversarial training | $1.82 \pm 0.04$ |
| ATRM | $\mathbf{1.56} \pm 0.05$ |

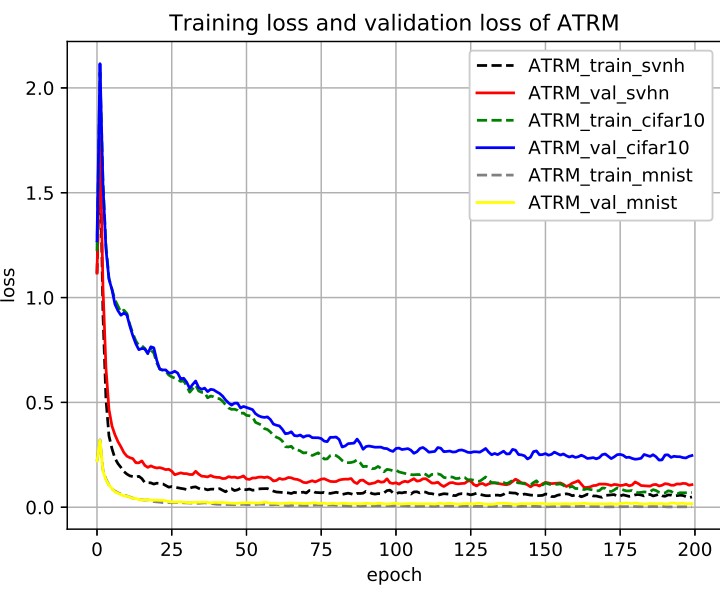

Figure 3: The convergence curves on three datasets

Finally, we have also done the experiments to prove the convergence for our proposed method. We have plotted the convergence curves for both training set and validation set of all three datasets as Figure 3 showing.

# 4 CONCLUSION

We present the novel framework called Adversarial Training with Riemannian Manifold (ATRM) which generalizes the traditional adversarial training method to Riemannian space. For traditional adversarial training methods, the worst perturbation is often searched with the gradient $\nabla_x \mathcal{L}$. However, when the data lay on a geometric manifold defined as a Riemannian manifold, the gradient $\nabla_x \mathcal{L}$ is not the steepest direction, leading the adversarial perturbation is not the worst one. We present a series of theory showing that our method leads to the steepest direction of the loss function in Riemannian space. In practice, we also develop a practical algorithm guaranteeing the decent direction for the loss function at each epoch. Experiments demonstrate encouraging results on benchmark data including MNIST, CIFAR-10 and SVHN.

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

## A    FIND THE ADVERSARIAL PERTURBATION WITHIN $l_2$-BALL ON MANIFOLD

Recall our goal is to maximize the value of the problem:

$$\arg\max_{\epsilon} \ \mathcal{L}(x) + \nabla_x \mathcal{L}^T \epsilon$$
$$s.t. \qquad g_{ij}(x)\epsilon^i \epsilon^j \leq \sigma^2$$

Since $\mathcal{L}(x)$ is independent of $\epsilon$ and the worst perturbation has the norm $\sigma$, then we have:

$$\arg\max_{\epsilon} \nabla_x \mathcal{L}^T \epsilon$$
$$\text{s.t.} \quad g_{ij}(x)\epsilon^i \epsilon^j = \sigma^2$$

Then we apply the Lagrangian multiplier method on this problem:

$$\arg\max_{\epsilon} \nabla_x \mathcal{L}^T \epsilon - \lambda(g_{ij}(x)\epsilon^i \epsilon^j - \sigma^2) \tag{21}$$

Making the first derivative of (21) with respect to $\epsilon$ zero, we have:

$$\nabla_x \mathcal{L} = \lambda g_{ij}(x)\epsilon^i$$

Then we have:

$$\epsilon = \frac{1}{\lambda} G^{-1} \nabla_x \mathcal{L}$$

Therefore:

$$\epsilon \propto G^{-1} \nabla_x \mathcal{L}$$

## B  FIND THE ADVERSARIAL PERTURBATION WITH $l_p$-BALL ON MANIFOLD

The optimization problem is:

$$\epsilon = \arg\max_{\epsilon} \ \mathcal{L}(x) + \nabla_x \mathcal{L}^T \epsilon$$
$$\text{s.t.} \quad \|\epsilon^T S\|_p \leq \sigma$$

Similar to Appendix **A**, we reduce the problem to:

$$\epsilon = \arg\max_{\epsilon} \ \mathcal{L}(x) + \nabla_x \mathcal{L}^T \epsilon$$
$$\text{s.t.} \quad \|\epsilon^T S\|_p = \sigma$$

We solve it with the Lagrangian multiplier method again and set $r = \epsilon^T S$ and $f(r) \equiv \|r\|_p = \sigma$. We have

$$\nabla_x \mathcal{L} r S^{-1} = \lambda(f(r) - \sigma)$$

Then we make the first derivative respect to $r$:

$$\nabla_x \mathcal{L} S^{-1} = \lambda \frac{r^{p-1}}{p(\sum_i \epsilon_i^p)^{1-\frac{1}{p}}}$$

$$\nabla_x \mathcal{L} S^{-1} = \frac{\lambda}{p}(\frac{r}{\sigma})^{p-1}$$

$$(\nabla_x \mathcal{L} S^{-1})^{\frac{p}{p-1}} = (\frac{\lambda}{p})^{\frac{p}{p-1}}(\frac{r}{\sigma})^p \tag{22}$$

If we sum over two sides, we have

$$\sum(\nabla_x \mathcal{L} S^{-1})^{\frac{p}{p-1}} = \sum(\frac{\lambda}{p})^{\frac{p}{p-1}}(\frac{r}{\sigma})^p$$

$$\|\nabla_x \mathcal{L} S^{-1}\|_{p^*}^{p^*} = (\frac{\lambda}{p})^{p^*} * 1$$

$$(\frac{\lambda}{p}) = \|\nabla_x \mathcal{L} S^{-1}\|_{p^*} \tag{23}$$

By combining (22) and (23), we have

$$r = \sigma sign(\nabla \mathcal{L}^T S^{-1})(\frac{|\nabla \mathcal{L}^T S^{-1}|}{\|\nabla \mathcal{L}^T S^{-1}\|_{p^*}})^{\frac{1}{p-1}}$$

Since $r = \epsilon^T S$, we have

$$\epsilon = \sigma sign(\nabla \mathcal{L}^T S^{-1})(\frac{|\nabla \mathcal{L}^T S^{-1}|}{\|\nabla \mathcal{L}^T S^{-1}\|_{p^*}})^{\frac{1}{p-1}} S^{-1}$$

## C  FIND THE APPROXIMATION FOR $|H|^{-1}\nabla_x \mathcal{L}(y_i, x_i, \theta)$.

We develop a modified BFGS method to approximate the $|H|^{-1}$. To solve the memory problem, we use the simplified Limited-memory BFGS (L-BFGS) to approximate $|H|^{-1}\nabla_x \mathcal{L}(y_i, x_i, \theta)$ directly (based on BFGS).

We first present Theorem C.1.

**Theorem C.1.** *(Sherman Morrison formula (Bartlett, 1951)) Let $A \in \mathbb{R}^{n \times n}$ be an invertible square matrix and $v, u \in \mathbb{R}^n$ be column vectors. Then $A + uv^T$ is invertible if and only if $1 + v^T A^{-1} u \neq 0$. And its inverse is given by*

$$(A + uv^T)^{-1} = A^{-1} - \frac{A^{-1}uv^T A^{-1}}{1 + v^T A^{-1}u} \tag{24}$$

We then take the second order Taylor expansion for $\mathcal{L}(x + \epsilon_1)$ (where $\epsilon_1$ is a small value):

$$\mathcal{L}(x + \epsilon_1) \approx \mathcal{L}(x) + \nabla \mathcal{L}(x)\epsilon_1 + \epsilon_1^T \nabla^2 \mathcal{L}(x)\epsilon_1$$

If we make the first derivative on both sides with respect to $\epsilon_1$, we can have:

$$\nabla \mathcal{L}(x + \epsilon_1) - \nabla \mathcal{L}(x) \approx H\epsilon_1$$

where $H$ denotes the Hessian matrix of $\mathcal{L}(x)$ with respect to $\epsilon_1$.

We can define $y = \nabla \mathcal{L}(x + \epsilon_1) - \nabla \mathcal{L}(x)$, then

$$y \approx H\epsilon_1 \tag{25}$$

Next, we use modified BFGS algorithm to approximate $H$ and let

$$B = I + \Delta M \tag{26}$$

where $B$ represents the approximation for $H$, $I$ is identity matrix, and $\Delta M$ denotes difference matrix. Our aim is to evaluate $\Delta M$. First we define

$$\Delta M = auu^T + bvv^T \tag{27}$$

where $a, b \in \mathbb{R}$ and $u, v \in \mathbb{R}^N$ are undetermined. It is easy to get that $\Delta M$ is symmetric. We combine (27), (26) and (25):

$$
\begin{aligned}
y &= I\epsilon_1 + auu^T\epsilon_1 + bvv^T\epsilon_1 \\
&= I\epsilon_1 + (au^T\epsilon_1)u + (bv^T\epsilon_1)v
\end{aligned}
\tag{28}
$$

We can further assume

$$au^T\epsilon_1 = 1, \quad bv^T\epsilon_1 = -1 \tag{29}$$

Therefore, we have

$$a = \frac{1}{u^T\epsilon_1}, \quad b = -\frac{1}{v^T\epsilon_1} \tag{30}$$

If we combine (28) and (30), we have

$$u - v = y - I\epsilon_1 \tag{31}$$

Therefore, we can set

$$u = y, \quad v = I\epsilon_1 \tag{32}$$

By combining (32) and (30), we have

$$a = \frac{1}{y^T\epsilon_1}, \quad b = -\frac{1}{\epsilon_1^T I\epsilon_1} \tag{33}$$

Then, we combine (26), (32), and (33):

$$B = I + \frac{yy^T}{y^T\epsilon_1} - \frac{\epsilon_1\epsilon_1^T}{\epsilon_1^T\epsilon_1} \tag{34}$$

(34) can be viewed as a simplified BFGS method with one step approximation. When the dimension of $s$ and $\epsilon_1$ increases, a large amount of memory is needed to store the matrix $ss^T$ and $\epsilon_1\epsilon_1^T$. To solve this problem, we then use the simplified L-BFGS method to evaluate directly $|H|^{-1}\nabla_x\mathcal{L}(y_i, x_i, \theta)$. We first use Theorem C.1 to reformulate (34) as:

$$D = (I - \frac{\epsilon_1 y^T}{y^T\epsilon_1})(I - \frac{y\epsilon_1^T}{y^T\epsilon_1}) + \frac{\epsilon_1\epsilon_1^T}{y^T\epsilon_1}$$

Then, we can easily implement the method L-BFGS as (Byrd et al., 1994) and get Algorithm 3.

## D  PROOF FOR LEMMA 2.2

Let $\xi(0)$ and $\xi(a)$ be two closed points on Riemannian manifold, where $\xi(t)$ is a shortest curve connecting these two points. Then, we have the square of geodesic distance between these two points:

$$ds^2 = g_{ij}(t)d\theta^i d\theta^j \tag{35}$$

where $d\theta^i = \dot{\xi}(t)^i$ is the $i^{th}$ component of small vector $d\theta$ and $g_{ij}(t)$ is the metric tensor.

---

**Algorithm 3** Approximation for $|H|^{-1}\nabla_x\mathcal{L}(y_i, x_i, \theta)$.

1: **function** APPROHD($y_i, x_i, \theta, \zeta$)
2:     Set $\zeta$ to a very small value
3:     $g_0 = \nabla_x\mathcal{L}(y_i, x_i, \theta)$
4:     $g_1 = \nabla_x\mathcal{L}(y_i, x_i + \zeta g_0, \theta)$
5:     $y = g_1 - g_0$
6:     $s = \zeta g_0$
7:     $\rho = \frac{1}{y^T s}$
8:     $\alpha = \rho s^T g_1$
9:     $q = g_1 - \alpha y$
10:     $r_0 = q_0$
11:     $\beta = \rho y^T r_0$
12:     $r_1 = r_0 + (\alpha - \beta)s$
13:     **return** $r_1$
14: **end function**

---

*Proof.* We evaluate the geodesic distance between $\xi(0)$ and $\xi(a)$ through Theorem 2.1:

$$d_s = \int_0^a \sqrt{g_{ij}(t)\dot{\xi}^i(t)\dot{\xi}^j(t)} \ dt \tag{36}$$

Since $\xi(0)$ and $\xi(a)$ are closed points, $a \to 0$. Then we have

$$d_s = \lim_{a \to 0} \int_0^a \sqrt{g_{ij}(t)\dot{\xi}^i(t)\dot{\xi}^j(t)} \ dt = \sqrt{g_{ij}(t)\dot{\xi}^i(t)\dot{\xi}^j(t)} \tag{37}$$

Therefore, we have

$$ds^2 = g_{ij}(t)d\theta^i d\theta^j \tag{38}$$

$\square$

## E  PROOF FOR LEMMA 2.3

Assume $H$ be a symmetric square matrix in $\mathbb{R}^n \times \mathbb{R}^n$ and $r \in \mathbb{R}^n$ be a vector. Then we have $|r^T H r| \leq r^T |H| r$ and $|H|$ represents the matrix with taking the absolute value of each eigenvalue of $H$.

*Proof.* Assume $\{e_1, e_2, ..., e_n\}$ and $\{\lambda_1, \lambda_2, ..., \lambda_n\}$ are the eigenvectors and corresponding eigenvalues of matrix $H$. Then we reformulate $|r^T H r|$ with eigenvectors and corresponding eigenvalues as:

$$|r^T H r| = |\sum_i (r^T e_i)\lambda_i(e_i^T r)| = |\sum_i \lambda_i(r^T e_i)^2| \tag{39}$$

We can now use the triangle inequality $|\sum_i r_i| \leq \sum_i |r_i|$ and we have:

$$|r^T H r| \leq \sum_i |\lambda_i(r^T e_i)^2| = \sum_i (r^T e_i)|\lambda|(r^T e_i) = r^T |H| r \tag{40}$$

$\square$

## F  PROOF FOR LEMMA 2.4

Let $A$ be a real symmetric positive definite matrix in $\mathbb{R}^n \times \mathbb{R}^n$. Then we have a unique positive definite matrix $S$ in $\mathbb{R}^n \times \mathbb{R}^n$ so that $A = S^2$.

*Proof.* To prove existence: Since $A$ is a real symmetric positive definite matrix, we have $A = P^T diag(\lambda_1, ..., \lambda_n)P$, where $P$ is an orthogonal matrix and $\{\lambda_i\}$ are the Eigen values of $A$ ($\lambda_i > 0$). We can find $S = P^T diag(\lambda_1^{\frac{1}{2}}, ..., \lambda_n^{\frac{1}{2}})$ that $A = S^2$.

To prove uniqueness: Let $B$ be another positive definite matrix and $A = B^2$. Since $B$ is positive definite, we have $A = T^T diag(\mu_1, ..., \mu_n)T$, where $T$ is an orthogonal matrix and $\{\mu_i\}$ are Eigen values of $B$ ($\mu_i > 0$). We have $A = P^T diag(\lambda_1, ..., \lambda_n)P$, therefore

$$T^{-1}diag(\mu_1, ..., \mu_n)T = P^{-1}diag(\lambda_1, ..., \lambda_n)P \tag{41}$$

Let $U = (u_{ij})_{n \times n} = PT^{-1}$ and we have:

$$diag(\mu_1^2, ..., \mu_n^2)U = Udiag(\lambda_1, ..., \lambda_n) \tag{42}$$

which is equivalent to:

$$\lambda_i u_{ij} = u_{ij}\mu_j^2 \tag{43}$$

When $\lambda \neq \mu_j^2$, $u_{ij} = 0$, we have $\lambda_i^{\frac{1}{2}} = u_{ij}\mu_j$. When $\lambda_i = \mu_j^2$, we also have $\lambda_i^{\frac{1}{2}} = u_{ij}\mu_j$. Therefore:

$$diag(\mu_1, ..., \mu_n)U = Udiag(\lambda_1^{\frac{1}{2}}, ..., \lambda_n^{\frac{1}{2}}) \tag{44}$$

Then we have:

$$B = T^{-1}diag(\mu_1, ..., \mu_n)T = P^{-1}diag(\lambda_1^{\frac{1}{2}}, ..., \lambda_n^{\frac{1}{2}})P = S \tag{45}$$

$\square$

