# OpenReview forum: "LEARNING ADVERSARIAL EXAMPLES WITH RIEMANNIAN GEOMETRY"
_ICLR.cc/2019/Conference_

### Official Review · AnonReviewer1 · 2018-10-29
**A weak submission, the Riemannian analysis seems flawed.**

**Rating:** 3
**Confidence:** 5

**Review:**

The authors argue that they propose a method to find adversarial examples, when the data lie on a Riemannian manifold. In particular, they derive a perturbation, which is argued to be the worst perturbation for generating an adversarial example, compared to the classical Euclidean derivation.

I strongly disagree with the proposed (Riemannian) geometric analysis, because there are several technical mistakes, a lot of arbitrary considerations, and flawed assumptions. In particular, my understanding is that the proposed method is not related at all with Riemannian geometry. For justification I will comment some parts:

#1) In Section 1 paragraph 4, and in Section 2.3 after Eq. 14, the sentences about the gradient of a function that is defined on a manifold are strange and unclear. In general, the gradient of a function defined on a manifold, points to the ascent direction. Thus, if I understood correctly the sentences in the paper support that the gradient of such a function is meaningless, so I think that they are wrong.

#2) How the $\ell_2$-ball on a manifold is defined? Usually, we consider a ball on the tangent space, since this is the only Euclidean space related to the manifold. Here, my understanding is that the authors consider the ball directly on the manifold. This is clearly wrong and undefined.

#3) To find the geodesic you have to solve a system of 2nd order non-linear ODEs, and there are additional details which I will not include  here, but can be easily found in the Riemannian geometry literature. Also, I think that the Lemma 2.2. is wrong, since the correct quantity of Eq. 3 is $ds^2 = g_ij(t)d\theta^i d\theta^j dt^2$, where $dt\rightarrow 0$ based on the included proof. This is clearly not a sensible geodesic, it is just the infinitesimal length of a line segment when $t\rightarrow 0$, which means that the two points are infinitesimally close.

#4) If $x, y$ is on a Riemannian manifold then the $x+y$ operator does not make sense, so Eq. 7 is wrong. In particular, for operations on Riemannian manifolds you need to use the exponential and the logarithmic map.

#5) Continuing from #4). Even if we consider the perturbation to be sufficiently small, still the $x+\epsilon$ is not defined. In addition, the constraint in Eq. 8 is wrong, because the inner product related to the Riemannian metric has to be between tangent vectors. Here the $\epsilon$ is an arbitrary quantity, since it is not defined where it actually lies. In general, the derivation here is particularly confusing and not clear at all. In my understanding the constraint of Eq. 8 is a purely linear term, and $d$ is not the geodesic distance on a manifold. It just represents the Mahanalobis distance between the points $x$ and $x+\epsilon$, for a matrix $G$ defined for each $x$, so it is a linear quantity. So Eq. 9 just utilizes a precondiner matrix for the classical linear gradient.

#6) The Eq. 12 is very flawed, since it equalizes a distance with the Taylor approximation error. I think that this is an unrealistic assumption, since these terms measure totally different quantities. Especially, if $d$ is the geodesic distance.

#7) The upper bound in inequality Eq. 13 comes from Eq. 12, and it is basically an assumption for the largest absolute value of the Hessian's eigenvalues. However, this is not discussed in the text, which are the implications?

#8) I find the paper poorly written, and in general, it lacks of clarity. In addition, the technical inconsistencies makes the paper really hard to follow and to be understood. I mentioned above only some of them. Also, there are several places where the sentences do not make sense (see #1), and the assumptions made are really arbitrary (see #6). The algorithms are not easy to follow. Minor comments, you can reduce the white spaces by putting inline Eq. 3, 4, 5, 6, 9, 10, 14, and Figure 2. The notation is very inconsistent, since it is very unclear in which domain/space each quantity/variable lies. Also, in Section 2.5. the authors even change their notation.

In my opinion, the geometrical analysis and the interpretation as a Riemannian manifold is obviously misleading. Apart from the previously mentioned mistakes/comments, I think that the proposed approach is purely linear. Since actually the Eq. 14 implies that the linear gradient of the loss function, is just preconditioned with the Hessian matrix of the loss function with respect to the input $x$. Of course, if this function is convex around $x$, then this quantity is the steepest ascent direction of the loss function, simply on the Euclidean space where $x$ lies. However, when this function is not convex, I am not sure what is the behavior when all the eigenvalues of the Hessian are set to their absolute value. Also, the (arbitrary) constraint in Eq. 13 implicitly sets a bound to the eigenvalues of the Hessian, which in some sense regularizes the curvature of the loss function. To put it simple, I think that the proposed method, is just a way to find a preconditioner matrix for the linear gradient of the loss function, which points to the steepest direction. This preconditioner is based on the Hessian of the loss function, where the absolute values of the eigenvalues are used, and also, are constrained to be bounded from above based on a given value.

Generally, in my opinion the authors should definitely avoid the Riemannian manifold consideration. I believe that they should change their perspective, and consider the method simply as a local preconditioner, which is based on the Hessian and a bound to its (absolute) eigenvalues. They should also discuss what is the implication by setting the eigenvalues to their absolute values. However, since I am not an expert in the field of adversarial examples, I am not sure how novel is this approach.

---

> ### Author Response · Authors · 2018-11-27
> **Response to reviewer 1**
>
> It is true that we eventually engage a preconditioner matrix for the linear gradient of the loss function to search the adversarial perturbation in this paper. However, we show that this preconditioner matrix is merely one special metric tensor definition (associated with one Riemannian manifold defined over the loss function). From the perspective of Riemannian geometry, we could define many different metric tensors (e.g., Fisher information metric), depending on different real scenarios; this offers us a generalized and new insight to study the adversarial examples. As one promising future direction, one can study the effects of different metric tensors and their physical meanings on adversarial perturbation.
>
> To Q1:
> We regret that the sentence caused a confusion. The gradient is not defined as the ascent direction on the manifold. We define a $d$-dimensional Riemannian space where all the data sit. Our aim is to find the direction along which we move a point $x$ with a small distance in the manifold so that the loss function can be increased the biggest ($L(x+\epsilon, \theta)-L(x, \theta)$).  For example, consider a 2-dimensional Riemannian manifold defined by a surface (within the 3-dimensional Euclidean space $R^3$), $r(u, v)=(g_1(u, v), g_2(u, v), g_3(u, v))$ defined with the metric tensor $G(u, v)$. Let the input be $x=(u, v)$ and $L(x, \theta)$ be the loss function of DNN. Then the steepest direction for Loss function on this surface is $G^-{1} \nabla_{x} L$.
>
> To Q2:
> Mathematically strictly speaking, it would be common and basic that a ball with an arbitrary radius can be defined given a specific space and a corresponding metric [1]. Specifically, we define in this paper a ball in the Riemannian space with the $l_p$ metric. We can easily define the ball on the manifold. Again, consider the 2-dimensional manifold surface $r(u, v)=(g_1(u, v), g_2(u, v), g_3(u, v))$ is defined with metric tensor G(u, v) and Let $x_1=(u_1, v_1)$ be a point on the surface with parametrization (u, v) and let $x_2=(u_1+\epsilon_1, v_1+\epsilon_2)$ be another point. When $\epsilon=(\epsilon_1, \epsilon_2)$ is very small, the distance between these two points can be approximate by $\epsilon G \epsilon^T$ and a small ball with center (u_1, v_1) can be defined as $\epsilon G(u_1, v_1) \epsilon^T<=\sigma^2$, where \sigma is small.
>
> To Q3:
> Thanks for pointing out this mistake we made carelessly. It should be $dot{\xi}^i dt$. It is indeed not rigorous to treat the first fundamental form as the geodesic distance. However, the distance between two close points can be computed as stated in (3). Hence, Lemma 2.2 is in general no problem though it was not rigorously stated in the submission. We have now revised this lemma to make it more precise. Details can be seen in the revised version.
> To Q4 and Q5:
> Again we think the reviewer might misunderstand the paper. To explain where $x$ and $x+\epsilon$ lie on, we again take the example: the surface $r(u, v)=(g_1(u, v), g_2(u, v), g_3(u, v))$ is in R^3 with metric tensor G(u, v) and Let $x_1=(u_1, v_1)$ be a point on the surface with parametrization (u, v) and let $x_2=(u_1+\epsilon_1, v_1+\epsilon_2)$ be another point. When $\epsilon=(\epsilon_1, \epsilon_2)$ is very small, the distance between these two points can be approximate by $\epsilon G \epsilon^T$.
>
> To Q6:
> We regret that the expression caused a confusion again. We have reformulated this in the revised paper. We also clarify it here. First, we derived in Section 2.2 an optimization problem in the general Riemannian manifold space that is exclusively decided by the metric tensor. In Section 2.3, we then defined a special metric tensor (or special Riemannian manifold space) based on the loss function. Such special metric tensor is reasonable for adversarial examples, since adversarial examples are closely related to the loss function and classification boundary. On the other hand, there are also other ways of defining a metric tensor (e.g., the famous fisher information matrix), which could be chosen in practice.
> For Q7.
> When the loss function is locally convex with respect to $x$, the Hessian matrix $H$ is positive semidefinite matrix and the absolute Hessian matrix is the same as the Hessian matrix.
> When the loss function is not locally convex, the non-negative eigenvalues of absolute Hessian matrix keep the same with the Hessian matrix while the negative eigenvalues are changed to positive ones. The curvature information is partially kept. On the other hand, the upper bound enables us an insight that the adversarial perturbation is along the adjusted gradient direction (adjusted by $|H|^{-1}$, which can be viewed as a metric tensor for a Riemannian manifold).
> For Q8.
> Follow some of your comments, we have proofread and revised the paper again.
> [1]. ‘Metric Spaces’,Stanisława Kanas, JOURNAL OF FORMALIZED MATHEMATICS

---

### Official Review · AnonReviewer2 · 2018-10-29
**Promising results but proposed framework is not general enough for a Riemannian manifold and seems wrong**

**Rating:** 4
**Confidence:** 5

**Review:**

1.	Some motivation of extending the adversarial examples generation on manifold should be there.
2.	Even if \epsilon is small, if x is on a manifold, x+\epsilon may not, so I am not sure about the validity of the definition in Eq. (7) and what follows from here. One solution is putting the constraint d(x, Exp_x(\epsilon)) \leq \sigma, which implies that g(\epsilon, \epsilon) \leq \sigma.
Also, x and \epsilon lies in completely different space, \epsilon should lie on the tangent space at x. So, I don’t understand why x+\epsilon makes sense? It makes the rest of the formulation invalid as well.
3.	I don’t understand why in Eq. (12), d(x, x+\epsilon)^2 = |m(x)|? Do authors want it to be equal, otherwise, I can not see why this equality is true.
4.	In Lemma 2.3, please make H in \mathbb{R}^{n\times n} instead of \mathbb{R}^n \times \mathbb{R}^n (same issue for Lemma 2.4), later does not make sense in this context. Also, why not write |H|=U|\Sigma|U^T, instead of what you have now.
5.	No need to prove Lemma 2.3 and 2.4. These are well-known results in matrix linear algebra.
6.	It’s nice that the authors generalize to l_p ball and can show FGSM as a special case.
7.	Some explanation of Algo. 2 should be there in the main paper given that it is a major contribution in the paper and also authors put a paper more than 8 pages long, so as a reader/ reviewer I want more detailed explanation in the main body.
8.	In Algorithm 1, step 7: “Update the parameters of neural network with stochastic gradient” should be updated in the negative direction of gradient.
9.	Algorithm 2 is clearly data driven. So, can authors comment on special cases of Algorithm 2 when we explicitly know the Riemannian metric tensor, e.g., when data is on hypersphere.
10.	Can authors comment on the contemporary work https://arxiv.org/pdf/1807.05832.pdf, as the purpose is very similar.
11.	The experimental validation is nice and showed usefulness of the proposed method.


Pros:
1.	A framework to show the usefulness of non-Euclidean geometry, specifically curvature for adversarial learning.
2.	Nice set of experimental validation.

Cons:
1.	Some theorems statement can be ignored to save space, e.g., Lemma 2.3 and 2.4. And instead, need some explanation of Algorithm 2 in the main text. Right now, not enough justification of additional page.
2.	Not sure about the validity of the main formulation, Eq. (7) and other respective frameworks when data x is on a manifold.

Minor comments:
1.	In page 2, “In this case, the Euclidean metric would be not rational.”-> “In this case, the Euclidean metric would not be rational”.
2.	 “However, in a geometric manifold, particularly in Riemannian space, the gradient of a loss function unnecessarily presents the steepest direction.” Not sure what authors meant by “unnecessarily presents”
3.	No need to reprove Lemma 2.2, just give reference to a differential geometry textbook like Chavel or Boothby.

I want the authors to specifically address the cons.

---

> ### Author Response · Authors · 2018-11-27
> **Response to reviewer 2**
>
> We define the Riemannian manifold with the dimension $d$, where all the data lie on it. Here the low-dimensional $d$ is not defined in $R^d$ but in the Riemannian manifold space. Under our assumption, both the natural example $x$ and perturbed example $x+\epsilon$ are in the $d$-dimensional Riemannian space (not $R^d$ again). One example can be seen in the response to Q9.
> When the perturbation is small enough, we can approximate the geodesic distance between $x$ and $x+\epsilon$ by $\epsilon G \epsilon^T$. Then we can define the ball with $l_p$ metric on the Riemannian space as (15) in the paper. Different from the Euclidean space, we define the ball with the Riemannian metric. (The ball in Euclidean is defined by the Euclidean metric). Our aim is to find the direction in which we move a point $x$ on a $d$ dimensional manifold with a small constant distance, which can enlarge the loss function the biggest.
>
> To Q1:
> On one hand, the Riemannian space is a generalized case of Euclidean space. A study on the generalized case could offer us new perspectives on adversarial examples. As a matter of fact,
> it is quite often that data sit not in a Euclidean space but have a Riemannian metric structure. In these cases, the ordinary gradient does not lead to a steepest descent direction of the target function. Therefore, it would be important to extend the adversarial examples generation on manifold which we could adjust to find the steepest direction.
>
>
> To Q2:
> If natural data are on a low dimensional manifold in $R^d$, it is true that $x+\epsilon$ may be not on the manifold. However, in this paper, we assume all the data are in the $d$-dimensional Riemannian space. Therefore, the natural example $x$ and non-natural example $x+\epsilon$ are both on the Riemannian manifold.  Moreover, the ball on it can be defined by $d(x, x+\epsilon) \leq \sigma$.
>
> To Q3:
> We regret that the expression caused a confusion. We have reformulated this in the revised paper. We also clarify it here. First, we derived in Section 2.2 an optimization problem in the general Riemannian manifold space that is exclusively decided by the metric tensor. In Section 2.3, we then defined a special metric tensor (or special Riemannian manifold space) based on the loss function. Such special metric tensor is reasonable for adversarial examples, since adversarial examples are closely related to the loss function and classification boundary. On the other hand, there are also other ways of defining a metric tensor (e.g., the famous fisher information matrix), which could be chosen in practice.
>
> To Q4, 5, 8:
> Following your comments, these have been corrected in the revised submission.
>
> To Q7:
> More explanations about Algo.2 have been added in the revised submission.
>
> To Q9:
> Algorithm 2 presents the calculation for product of matrix and vector. In this paper, we construct the Riemannian manifold with the metric tensor defined by absolute Hessian matrix. Therefore, the worst perturbation is approximated as $|H|^{-1} \nabla L$. For an arbitrary manifold with metric tensor $G$, the steepest direction is $G^{-1} \nabla L$. For example, consider a 2-dimensional manifold defined by a surface (within the 3-dimensional Euclidean space $R^3$), $r(u, v)=(g_1(u, v), g_2(u, v), g_3(u, v))$ defined with the metric tensor $G(u, v)$. Let the input be $x=(u, v)$ and $L(x, \theta)$ be the loss function of DNN. Then the steepest direction for Loss function on this surface is $G^-{1} \nabla_{x} L$.
>
> To Q10:
> This paper has been well noted. Their purposes/motivations are quite different between the paper and ours. In the mentioned paper, the aim is to exploit the adversarial training to smooth the low dimensional statistic manifold. In this work, we focus on generating and resisting the adversarial examples through the perspective of the geometry. These two approaches could even be combined.

---

> > ### Comment · AnonReviewer2 · 2018-11-28
> > **Response**
> >
> > I thank the authors for trying to clarify some of the comments raised. But I still have some major unanswered claims such as:
> > 1) "If natural data are on a low dimensional manifold in $R^d$, it is true that $x+\epsilon$ may be not on the manifold. However, in this paper, we assume all the data are in the $d$-dimensional Riemannian space. Therefore, the natural example $x$ and non-natural example $x+\epsilon$ are both on the Riemannian manifold. " This does not make sense, because if all the data are in the $d$-dimensional Riemannian space, does not imply $x$ and $x+\epsilon$ lie on that space. And what authors meant by "non-natural"? I can not see the validity of Eq. (7) and the following equations.
> >
> > 2) "In the mentioned paper, the aim is to exploit the adversarial training to smooth the low dimensional statistic manifold. In this work, we focus on generating and resisting the adversarial examples through the perspective of the geometry." what the authors meant by "resisting the adversarial examples"? Also not clear what is the difference? To me, both papers are trying to exploit geometry to learn adversarial samples.
> >
> > 3) Going back to my earlier comment, $x+\epsilon$ does not make any sense as the summands lie on different spaces where "+" does not make sense. If x is in Euclidean space, then $x+\epsilon$ makes sense but then the entire purpose of this work becomes moot.
> >
> > Overall, though I appreciate the authors trying to clarify, I am still not sure about the validity of the main equations presented in this work. And I strongly believe that the equations presented are wrong.

---

> > > ### Author Response · Authors · 2018-11-29
> > > **Further clarification:**
> > >
> > > Thanks again for taking time in reviewing the paper and offer valuable comments. We further clarify the reviewer’s concerns as follows:
> > >  Response to 1) and 3):
> > >  We are afraid that Eq. (7) is no problem.
> > > For explaining that, we take an example: consider a 2-dimensional manifold defined by a surface (within the 3-dimensional Euclidean space $R^3$), $r(u, v)=(g_1(u, v), g_2(u, v), g_3(u, v))$ defined with the metric tensor $G(u, v)$. Let one point be $x=(u_1, v_1)$ and another point be $x+\epsilon=(u_1+\epsilon_1, v_1+\epsilon_2)$ where $\epsilon=(\epsilon_1, \epsilon_2)$. It is clear that $r(u_1, v_1)=(g_1(u_1, v_1), g_2(u_1, v_1), g_3(u_1, v_1))$ and $ r(u_1+\epsilon_1, v_1+\epsilon_2)=(g_1(u_1+\epsilon_1, v_1+\epsilon_2), g_2(u_1+\epsilon_1, v_1+\epsilon_2), g_3(u_1+\epsilon_1, v_1+\epsilon_2))$ are both on 2-dimensional manifold $r(u, v)$. It is also clear that no matter how $v$ and $u$ change, the point is always on manifold $r(u, v)$. When $\epsilon=(\epsilon_1, \epsilon_2)$ is very small, the distance between these two points can be approximated by $\epsilon G \epsilon^T$ as defined in (8).
> > > Please also refer to Page 282 of book [1] for the detailed explanation about this. In addition, there are a lot of related work in neural networks which tries to conduct the optimization on the Riemannian space [2][3]. We regret to cause you the confusion. We will add more texts to explain this in the paper.
> > >
> > > Response to 2):
> > > a) Explanation about the “resisting the adversarial examples”:  Adversarial examples are first noticed because most of the good deep learning models would misclassify these examples with a high confidence. The straightforward motivation is then how to design a good and robust deep neural network model that is able to classify the adversarial examples correctly. We call this is to enable the network a resistance to the adversarial examples.
> > > b) To further explain the difference between the mentioned paper and this paper: In the mentioned paper, the author models the low dimensional statistic manifold defined with GMM in the space of a specific hidden layer of DNN. They still try to find in the EUCLIDEAN SPACE the worst-case perturbation (or the adversarial example) that make the statistical manifold not smooth. In comparison, in our paper, we search in the RIEMANNIAN MANIFOLD space the worst-case perturbation (or the adversarial example) that maximizes a given loss function.
> > >
> > >
> > > [1] Information Geometry and Its Applications, Shun-ichi Amari, book.
> > > [2] Riemannian metrics for neural networks I: Feedforward networks, Yann Ollivier. A Journal of the IMA.
> > > [3] Natural gradient works efficiently in learning, Shun-ichi Amari, Neural computation.

---

> > > > ### Comment · AnonReviewer2 · 2018-11-29
> > > > **Response to the first clarification**
> > > >
> > > > Let me give you a counterexample to Eq. 7.
> > > > 1) d is the distance on the chosen manifold.
> > > > 2) So x and x+epsilon both should be on the manifold.
> > > > 3) Let the manifold be unit 2 sphere, i.e., S^2. Let x be (1,0,0)^t and epsilon be (0,0.001,0.002)^t. Then x+epsilon is (1,0.001,0.002)^t. The norm of x+epsilon is not 1, hence x+epsilon does not lie on S^2.
> > > >
> > > >
> > > > Please identify which step(s) is(are) wrong in the above example.

---

> > > > > ### Author Response · Authors · 2018-12-01
> > > > > **Further clarification**
> > > > >
> > > > > Thanks for your comments.
> > > > >
> > > > > In the  example that the reviewer metioned,  x and x+\epsilon is given in the 3-dimensional  Euclidean space, though its manifold is actually a 2-d sphere.  In this case, x+\epsilon is  indeed not on the manifold.  However, in our paper, we directly assume that x is in the 3-d Riemannian (or manifold) space (not in the Euclidean space). In this case, x+\epsilon must also be in the  Riemannian space given that \epsilon is a pertubation in the Riemannian space.  Note that the pertubation can also be done in a NN's certain hidden layer which is generally considered reasonable to be in a Riemannian space.  In your example，the distance between two close points x and x+\epsilon in the 3-d Euclidean space is \sqrt(\epsilon \epsilon^T). In comparion,  under our assumption, x and x+\epsilon are in a 3-d Riemannian space with distance \sqrt(\epsilon G \epsilon^T), where G is the metric tenor that can be defined in various ways.

---

> > > > > > ### Comment · AnonReviewer2 · 2018-12-01
> > > > > > **Response**
> > > > > >
> > > > > > ```"In the  example that the reviewer metioned,  x and x+\epsilon is given in the 3-dimensional  Euclidean space, though its manifold is actually a 2-d sphere." ---> No, here x and x+\epsilon both are on a 2 dimensional manifold. S^2 is embedded in R^3, but the dimension of the manifold is 2, not 3. Please clarify.

---

> > > > > > > ### Author Response · Authors · 2018-12-03
> > > > > > > **Further clarification**
> > > > > > >
> > > > > > > We are afraid that the reviewer misunderstood the idea in the paper or there are still certain confusions.
> > > > > > >
> > > > > > > Again, though the reviewer considers x and \epsilon are both on a 2 dimensional manifold, they are represented with the coordinates in the 3-d Euclidean space.  In this case, x+\epsilon is of course not necessarily on the 2-dimensional manifold, since the "+" is conducted in the 3-d Euclidean space rather than the low-dimensional manifold or a Riemannian space we defined.
> > > > > > >
> > > > > > > It is emphasized again that we assume the data x are already in a Riemannian space, which is defined by a metric tensor G (the shape of G is 3*3). The perturbation $epsilon$ is also in the Riemannian space. x+\epsilon must also be in the Riemannian space. Importantly, we should note that the Riemannian space is different from the low-dimensional manifold space specified in your example. Actually, we don’t try to get the explicit low-dimensional manifold space which we believe is not straightforward to find especially when the data is very high-dimensional. We assume the data are in a certain Riemannian space (3-d manifold not 2-d) defined by G and we'd just try to reasonably measure the distance between two points associated with the Riemannian space (defined by a reasonable G with physical meanings).
> > > > > > >
> > > > > > > Back to your example again, this 3-d space is still the Euclidean space. The distance between two points x and x+\epsilon is still calculated as \sqrt(\epsilon^T\epsilon). In comparison, we think that the distance should be measured by \sqrt(\epsilon^TG \epsilon), where G, associated with a Riemannian space, represents the metric tensor and can be defined in many ways. This G is different from the low-dimensional manifold (i.e., the 2-d sphere in the example), which is difficult to find in practice.  In this paper, we try to define G from the perspective of the loss function. We believe this is reasonable, since the worst-case perturbation, i.e., the adversarial example, is mathematically designed to make worse (or maximize) the loss function. Experimental results in the paper also verify that such a modification with this specific G could achieve the state-of-the art performance.
> > > > > > >
> > > > > > > Following your possible thoughts, one idea is perhaps like the following: from the raw data x (in R^d), a low-dimensional manifold z (in S^t) is sought, where the G (of size tXt) can be obtained. All the data can be firstly transformed to S^t. Then the distance between a pair of points z_1, z_2, in S^t can be measured by \sqrt((z_1-z^2)^TG(z_1-z^2)). This idea is no problem; however, it is again usually not straightforward to obtain the low-dimensional manifold from raw data especially the very high-dimensional data. Instead, we don’t try to find the low-dimensional manifold but take another alternative and just would like to define a reasonable G with certain physical meanings. To this end, there have been a lot of related work, e.g., engaging the fisher information matrix to define G. Future work on this direction can be investigated.
> > > > > > >
> > > > > > > We hope this could clarify the concerns.

---

> > > > > > > > ### Comment · AnonReviewer2 · 2018-12-03
> > > > > > > > **Response**
> > > > > > > >
> > > > > > > > The authors just cannot say $x$ and $x+\epsilon$ both lie on the same manifold. This is untrue for any non-flat spaces.
> > > > > > > >
> > > > > > > > My feeling is the authors might use wrong terminologies/ notations here. I can not justify the equations in the current paper as the way they are presented.

---

> > > > > > > > > ### Author Response · Authors · 2018-12-04
> > > > > > > > > **Response**
> > > > > > > > >
> > > > > > > > > We appreciate your time in responding the comments.
> > > > > > > > >
> > > > > > > > > Let us put in this way, consider $x$ is already the coordinate in a Riemannian space (once again we emphasize that this Riemannian space is of the same dimension of x and is different from the low-dimensional manifold)
> > > > > > > > > This is the basic assumption in the paper.
> > > > > > > > > We would also find the best small $\epsilon$, still coordinated  in this Riemannian space. We can't see why $x+\epsilon$ is not in the Riemannian space.
> > > > > > > > >
> > > > > > > > > We feel sorry if the notations or terminologyies used in this paper misled you.

---

> > > > > > > > > > ### Comment · AnonReviewer2 · 2018-12-04
> > > > > > > > > > **Response**
> > > > > > > > > >
> > > > > > > > > > Can authors please give an example of such a x and epsilon on S^2?

---

> > > > > > > > > > > ### Author Response · Authors · 2018-12-04
> > > > > > > > > > > **Response**
> > > > > > > > > > >
> > > > > > > > > > > If x_1=(u_1, v_1) and \epsilon=(\epsilon_1, \epsilon_2) are 2-dimensional. Assume that they are in a Riemannian space with non-identity metric tensor, i.e. a 2-d sphere embedded in R^3: r(u, v)=(sin(u)cos(v), sin(u)sin(v), cos(v)). Note that x_1 and \epsilon is the coordinate in the Riemannian space, instead of in R^3.
> > > > > > > > > > >
> > > > > > > > > > > For the 2-d sphere, it is straightforward that the point x_2=x_1+\epsilon : r(u_1+\epsilon_1,v_1+\epsilon_2)=(sin(u_1+\epsilon_1)cos(v_1+\epsilon_2),sin(u_1+\epsilon_1)sin(v_1+\epsilon_2), cos(v_1+\epsilon_2)) is still on the 2-d sphere r(u, v). When \epsilon is a small value, the distance between x_1 and x_2 can be computed as \epsilon^T G \epsilon, where G is the metric tensor of the sphere which can be easily computed.

---

> > > > > > > > > > > > ### Comment · AnonReviewer2 · 2018-12-04
> > > > > > > > > > > > **Response**
> > > > > > > > > > > >
> > > > > > > > > > > > Then none of $x_1$ or $\epsilon$ is on the manifold, but in a chart of the manifold. This is very much misleading. Nowhere it is clear in the paper. I suggest the authors to rewrite the paper as in it's current form it is very unclear.

---

> > > > > > > > > > > > > ### Author Response · Authors · 2018-12-05
> > > > > > > > > > > > > **Response**
> > > > > > > > > > > > >
> > > > > > > > > > > > > Again, we regret that the paper may not be presented in a clear way. We will further enhance it with more interpretations as well as illustrations.
> > > > > > > > > > > > >
> > > > > > > > > > > > > Nonetheless, please note such notation that $x$ and $x+epsilon$ are in the Riemannian space has been commonly used in the related machine learning studies on Riemannian geometry. Please refer to p282 [1] and [2][3]. We are confident that the mathematics are no problem in the paper though some clarification should be made in the paper. And more importantly, our paper presents the first study of the concepts of adversarial examples from the perspective of Riemannian geometry. We also construct an intrinsic Riemannian space based on the loss function, with a property that the descent direction can be maintained at each training step. Our experiments also validate that this new framework could achieve superior performance to the state-of-the-art adversarial example approaches.
> > > > > > > > > > > > >
> > > > > > > > > > > > > [1] Information Geometry and Its Applications, Shun-ichi Amari, book.
> > > > > > > > > > > > > [2] Riemannian metrics for neural networks I: Feedforward networks, Yann Ollivier. A Journal of the IMA.
> > > > > > > > > > > > > [3] Natural gradient works efficiently in learning, Shun-ichi Amari, Neural computation.

---

### Official Review · AnonReviewer3 · 2018-11-02
**Promising performance**

**Rating:** 6
**Confidence:** 2

**Review:**

In the paper, the authors proposed to solve the learning problem of adversarial examples from Riemannian geometry viewpoint. More specifically, the Euclidean metric in Eq.(7) is generated to the Riemannian metric (Eq.(8)). Later, the authors built the correspondence between the metric tensor and the higher order of Taylor expansions.  Experiments show the improvement over the state-of-the art methods.

Some questions:
First of all, the idea of introducing Riemannian geometry is appealing.
In the end, a neural network can be roughly viewed as a chart of certain Riemannian manifold.
The challenging part is how can you say something about the properties of the high dimensional manifold, such as curvature, genus, completeness etc.
Unfortunately, I didn't find very insightful analysis about the underlying structure.
Which means, hypothetically, without introducing Riemannian geometry we can still derive Eq.(14) from Eq.(12), Taylor expansion will do the work.
So more insights about metric tensor G determined manifold structure can be very helpful.

Second, Lagrange multipliers method is a necessary condition, which means the search directions guided by the constraint may not lead to the optimal solutions.
It would be better if the authors can provide either theoretical or experimental study showing certain level of direction search guarantee.

Last, the experiment results are good, though it lacks of detailed discussion, for example could you decompose the effect achieved by proposed new Riemannian constraint and neural network architecture? Merely demonstrating the performances does not tell the readers too much.

---

> ### Author Response · Authors · 2018-11-27
> **Response to reviewer 3**
>
> To Q1:
>
> Thanks for your comments. Our paper offers a new insight to study adversarial examples (from the perspective of Riemannian geometry). Our work is the first one that investigates the effect of the norm on adversarial examples. We also construct an intrinsic Riemannian space based on the loss function, with a property that the descent direction can be maintained at each training step. Our paper offers a new starting point which can inspire new insight to study adversarial examples. On the other hand, we agree that it is better to define the physical meaning of manifold properties like the curvature, which could be one of our future work. In the future, it is also meaningful to study how to define other metric tensors (like fisher information matrix).
>
>
> To Q2:
> We agree that the Lagrange multiplier is just the necessary condition. Similar to almost all the optimization associated with neural networks, the global optimum of the loss function could not be guaranteed with the proposed algorithm in the paper.  Nonetheless, we manage to find the steepest direction on the defined manifold. We prove that each training step would point to the descent direction, which could guarantee a local optimum. This proposed algorithm was verified to be effective with very promising empirical results. To better visualize the convergence property, we also added the plots of convergence curves in the revision.
>
> To Q3:
> We actually have already provided the performance of baseline model (CNN model), baseline + l_2 adversarial training, and our proposed method (i.e., baseline + l_2 Riemannian constraint adversarial training). Compare with these three methods, we can conclude that our proposed Riemannian constraint can improve the performance of CNN model and appears more appropriate than the adversarial constraint defined in the Euclidean space.

---

### Meta-Review · Area_Chair1 · 2018-12-17
**Technical correctness issues**

**Confidence:** 4
**Recommendation:** Reject

**Metareview:**

On the positive side, this is among the first papers to exploit non-Euclidean geometry, specifically curvature for adversarial learning. However, reviewers are largely in agreement that the technical correctness of this paper is unconvincing despite substantial technical exchanges with the authors.